# Towards Understanding Tumour Colonisation by Probiotic Bacterium *E. coli* Nissle 1917

**DOI:** 10.3390/cancers16172971

**Published:** 2024-08-26

**Authors:** Georgette A. Radford, Laura Vrbanac, Rebekah T. de Nys, Daniel L. Worthley, Josephine A. Wright, Jeff Hasty, Susan L. Woods

**Affiliations:** 1Adelaide Medical School, University of Adelaide, Adelaide, SA 5000, Australia; 2Precision Cancer Medicine, South Australian Health and Medical Research Institute (SAHMRI), Adelaide, SA 5000, Australia; 3Colonoscopy Clinic, Spring Hill, Brisbane, QLD 4000, Australia; 4Synthetic Biology Institute, University of California, San Diego, CA 92093, USA; 5Department of Bioengineering, University of California, San Diego, CA 92093, USA; 6Molecular Biology Section, Division of Biological Sciences, University of California, San Diego, CA 92093, USA; 7Center for Microbiome Innovation, University of California, San Diego, CA 92093, USA

**Keywords:** *E. coli* Nissle 1917 (EcN), bacteria, colonisation, tumour homing, genetic engineering

## Abstract

**Simple Summary:**

Tumour-homing bacteria present as an ideal chassis for cancer treatment and detection. A decade of pre-clinical and clinical research with the tumour-homing bacteria *Escherichia coli* Nissle 1917 (EcN) has demonstrated that genetic modification for payload delivery can lead to significant tumour regression and, more recently, tumour detection. Currently unknown in the field is a deep mechanistic understanding of why bacteria home to and colonise tumours. This review summarises existing literature to provide insight into the tumour-homing and colonising ability of EcN, in addition to exploring other tumour-homing strains and pathogenic bacteria for a more in-depth view. A mechanistic understanding of this capability could prove invaluable in the development of the next-generation EcN chassis for tumour detection and treatment, as well as address biosafety and containment concerns necessary for clinical translation.

**Abstract:**

The last decade has seen a rapid increase in studies utilising a genetically modified probiotic, *Escherichia coli* Nissle 1917 (EcN), as a chassis for cancer treatment and detection. This approach relies on the ability of EcN to home to and selectively colonise tumours over normal tissue, a characteristic common to some bacteria that is thought to result from the low-oxygen, nutrient-rich and immune-privileged niche the tumour provides. Pre-clinical studies have used genetically modified EcN to deliver therapeutic payloads that show efficacy in reducing tumour burden as a result of high-tumour and low off-target colonisation. Most recently, the EcN chassis has been expanded into an effective tumour-detection tool. These advances provide strong justification for the movement of genetically modified EcN into clinical oncology trials. What is currently unknown in the field is a deep mechanistic understanding of how EcN distributes to and localises within tumours. This review summarises the existing EcN literature, with the inclusion of research undertaken with other tumour-homing and pathogenic bacteria, to provide insights into possible mechanisms of EcN tumour homing for future validation. Understanding exactly how and why EcN colonises neoplastic tissue will inform the design and testing of the next generation of EcN chassis strains to address biosafety and containment concerns and optimise the detection and treatment of cancer.

## 1. Introduction

The use of bacteria for cancer treatment dates back to the early 1800s when the foundation was set for a now rapidly evolving field that aims to provide novel therapeutic options for cancer patients. By the late 1800s, William Coley was innovatively using two bacteria, *Streptococcus pyogenes* and *Bacillus prodigiosus*, as a cancer treatment for sarcoma patients after he witnessed tumour regression in patients with severe bacterial infections [1,2,3]. However, the practice was short-lived due to safety concerns surrounding the injection of these live pathogenic bacteria. The key exception was the use of a live, attenuated strain of *Mycobacterium bovis*, termed Bacillus Calmette–Guérin, that for over 40 years now has successfully been used in the treatment of bladder cancer [4,5]. Renewed focus on the use of bacteria for cancer treatment occurred in the mid-1990s as recombinant DNA technologies provided avenues to modulate pathogenic bacterial genes, and molecular characterisation of the tumour microenvironment identified new targets for cancer treatment. Current research in this area spans a variety of cancer types and bacterial species, with the most commonly utilised tumour-homing bacteria including *Escherichia* [6,7,8,9,10,11,12], *Clostridium* [13], *Salmonella* [14,15,16,17,18], and *Bifidobacterium* [19]. Of these, the probiotic strain *Escherichia coli* Nissle 1917 (EcN) remains a promising bacterial chassis for a wide variety of cancer types due to its translatability for clinical use and tumour-homing behaviour. This review summarises the growing body of research investigating the use of EcN for cancer treatment and detection within both the pre-clinical and clinical cancer fields. We focus on the ability of EcN to home to tumours and discuss our current understanding of EcN tumour localisation and potential mechanisms driving this behaviour to guide the future development of EcN as a tumour-seeking chassis for cancer detection and treatment.

## 2. Fit for Purpose: Inherent Properties of Probiotic *E. coli* Nissle 1917

The human gut is estimated to contain approximately 10^14^ bacteria, with the colon alone containing approximately 10^11^ bacteria per millilitre [20]. Included in this vast intestinal microbiome is the species *Escherichia coli*, which itself comprises approximately 700 serotypes. EcN is an established probiotic strain (serotype O6:K5:H1) sold under the trade name Mutaflor^TM^. Its anti-inflammatory properties are exploited in the clinical treatment of intestinal disorders, such as constipation, ulcerative colitis and Crohn’s disease, as reviewed [21]. To achieve such effects, continual dosing of EcN is necessary for effective gut colonisation. While some healthy adult mice and humans can maintain EcN colonisation for up to 24 weeks after treatment discontinuation, the majority clear EcN in 2 weeks [8,22,23,24].

To survive in the highly competitive gut environment, as reviewed by Sonnenborn and Schulze, EcN employs a range of fitness factors, including antimicrobial microcins, such as H47 and M, iron-scavenging mechanisms, including siderophores, secretion of biofilm matrix components such as cellulose and expression of adhesins [21]. These fitness properties enable EcN to withstand inflammation and outcompete other enteric pathogens, ultimately preventing inflammation [25,26] and maintaining intestinal tight junction integrity, potentially through the deployment of outer membrane vesicles, as shown in vitro [27]. EcN’s propulsive flagella enables motility, aids in adhesion and facilitates chemotactic homing [21]. Classed as a facultative anaerobe, EcN can survive in both aerobic and anaerobic conditions. Such attributes, favouring both movement and enhanced survival in varied conditions, allow EcN to successfully transit through the gastrointestinal tract as well as colonise tumours, as consistently observed in a variety of pre-clinical models [6,9,10,11,12,28,29,30,31,32,33,34,35,36,37,38,39,40,41,42]. 

EcN is inherently sensitive to serum as a result of a point mutation that affects the generation of the lipopolysaccharide coat (semi-rough, O6 antigen phenotype) [43]. This results in relatively quick clearance of EcN from systemic circulation [21], contributing to the high-tumour and low ‘off-target’ organ colonisation that we and others have observed in mouse cancer models [10,11,12,29] and more recently in a small clinical trial [8]. EcN was isolated in the pre-antibiotic era and is not resistant to any commonly used antibiotics that are useful against Gram-negative bacteria. Together, the ineffective colonisation of off-target tissues and sensitivity to serum and antibiotics are important safety features of the EcN chassis strain. 

EcN have fimbriae, or hair-like appendages, that enable the bacteria to adhere to intestinal epithelial cells [44]. While other tumour-homing bacteria, such as *Salmonella typhimurium,* are inherently intracellular species in mammalian hosts, most gut commensal *E. coli* strains (including EcN) remain extracellular [45], but they may also be internalised by epithelial cells at low levels or phagocytosed by host cells [46]. In vitro co-culture of EcN with dendritic cells resulted in EcN internalisation by the dendritic cells, likely via actin filament polymerisation and phagocytosis, as cytochalasin D inhibited this process [30]. A similar mechanism was suggested to occur in vivo with antigen-presenting cells internalising EcN in intra-tumourally treated subcutaneous melanoma and lymphoma models [30]. A mouse breast carcinoma study using EcN minicells, the nanosized bacteria that bud off the parental bacteria but lack parental chromosomal DNA, also enabled visualisation of intracellular EcN minicells following intravenous administration [28]. This suggested intact EcN may also be internalised by tumour cells in a similar manner. We have recently reported the localisation of EcN primarily on the luminal tumour surface of colorectal tumours following oral administration of EcN to an orthotopic mouse model [8]. When taken together, it is evident that further detailed studies are required to understand how EcN attach and/or are taken up intracellularly by tumour cells. It is important to understand the intra- and/or extracellular localisation of EcN to inform the design and delivery of cancer therapeutics, to allow for both extracellular and intracellular targeting.

Unmodified EcN has a long history of use in humans [47]. The current day well-mapped, complete EcN genome, with key probiotic genes and their products identified, has enabled advanced genetic engineering of the EcN chassis for cancer research. This has led to the development of strains with improved stability and tumour targeting, for example bacterial encapsulation to temporarily evade immune clearance [33]. Given the promising clinical translatability of tumour homing, EcN as a therapeutic chassis, the last decade has seen an exponential increase in pre-clinical research utilising genetically engineered EcN for cancer treatment (Figure 1). 

## 3. Anti-Cancer Activity of Genetically Engineered EcN in Pre-Clinical Tumour Models

A common modification has been to genetically engineer EcN to produce pro-apoptotic proteins (such as sulforaphane, azurin and theta toxins) [12,31,32,33,37,42]. Intravenous, intraperitoneal, and oral administration of these engineered EcN to mouse models of colitis-induced colorectal cancer, orthotopic breast cancer, subcutaneous melanoma, colorectal and hepatocellular carcinoma resulted in significantly decreased tumour burden [12,31,32,33,37,42]. Furthermore, EcN genetically engineered to produce or carry cytotoxic compounds reduced tumour growth in mouse models of breast, colorectal and head and neck squamous cell cancers following single or multiple intravenous or intraperitoneal dose(s) [9,11,36,51,52]. Likewise, EcN engineered to synthesise 5-ALA, an amino acid that can be metabolised into a photosensitiser by tumour cells and excited to produce reactive oxygen species, resulted in significant growth reduction of xenografted colorectal tumours in a mouse model following intraperitoneal dosing [7]. EcN has also been modified to express and release immune-activating payloads, including chemokine ligand 16, cytokine granulocyte-macrophage colony-stimulating factor, the bacterial messenger cyclic di-AMP, nanobodies against CD47 or programmed cell death–ligand 1 and cytotoxic T lymphocyte-associated protein-4 [30,34,40,53]. Treatment with these EcN strains resulted in immune cell tumour recruitment and subsequent establishment of immunological memory, ultimately slowing tumour growth and increasing mouse survival using subcutaneous models of syngeneic lymphoma, melanoma, colorectal and breast cancer [30,34,40,53]. Together, this research spanning the last decade highlights the versatility and potential for engineered EcN strains as cancer therapeutics.

Recently, engineered EcN has also been used for tumour detection by use of strains that form gas vesicles, visible by non-linear ultrasound signals [54,55] or that release urine-detectable metabolites [8,35]. Both methods resulted in significant detection of neoplasia over controls using mouse models of orthotopic intestinal precancerous polyps, xenografted colorectal cancer, lymphoma, and liver metastases after oral EcN delivery [8,35,54,55]. 

In conclusion, engineered EcN strains demonstrate great potential in pre-clinical studies as both a cancer detector and therapeutic. This research encompasses a substantial array of cancer types all utilising the tumour-specific homing behaviour of EcN. When considering the translatability of these strategies into clinical practice, large questions remain about the exact mechanisms controlling this trait. Understanding this tumour-specific homing will help to optimise bacterial strains for use and may provide biocontainment strategies critical for trial in humans. 

## 4. First in Human Testing of Engineered EcN in Cancer Patients: Points for Consideration

To date, only a handful of clinical trials have used genetically modified EcN, with the majority focused on the probiotic functions of unmodified EcN. Recently, we reported the enrichment of non-modified EcN in tumour samples from colorectal cancer patients in comparison to matched normal tissue after oral dosing for two weeks, suggesting EcN tumour-homing behaviour may be conserved in humans [8]. Synlogic, an American-based synthetic biology company, has paved the way for the clinical translation of genetically modified EcN. The most relevant Synlogic clinical trial to the present discussion utilised SYNB1891, an EcN strain that produces cyclic di-AMP under hypoxic conditions, such as is found in the tumour microenvironment. SYNB1891 was first delivered intra-tumourally to mouse models of lymphoma and solid tumours, which, upon phagocytosis by antigen-presenting cells, initiated an intracellular stimulator of the interferon (STING) pathway [30,56]. Downstream production of Type I interferons (IFN) and increased inflammatory cytokine production facilitated anti-tumour immunity, resulting in tumour regression [30]. The subsequent Phase I trial recruited 32 advanced, treatment-refractory cancer patients and included both a dose-escalation arm, where patients received multiple percutaneous, intra-tumoural doses of SYNB1891 alone, and a combination treatment arm, with SYNB1891 and anti-PD-L1 inhibitor Atezolizumab (NCT04167137) [56]. Participant retention in the trial was low, with only 13% (4 participants) receiving the full four treatment cycles, with trial withdrawal primarily due to disease progression [56]. The trial was also closed early by Synlogic to focus on the development of non-cancer leads [56]. Despite this, the safety of this approach was demonstrated, with only one patient experiencing a grade 3 serious adverse event and no deleterious SYNB1891 infections observed [56]. Four of twenty-five evaluable participants achieved stable disease for over 2 months in this heavily pre-treated population, and most exhibited activation of the STING pathway, as indicated by upregulation of IFN-stimulated genes in core tumour biopsies following SYNB1891 treatment [56]. 

The Synlogic approach involves intra-tumoural injection of SYNB1891 combined with hypoxia-regulated production of the STING agonist to maximise on-target delivery to the tumour and reduce off-target complications. This has limited this approach to accessible, percutaneous injectable tumours, resulting in the exclusion of some of the most common cancers that form in the gastrointestinal tract and lungs from the Phase I SYNB1891 trial. Meanwhile, alternate methods to enable tumour-specific release of genetically encoded payloads from bacteria have been developed using pre-clinical models, as reviewed [48]. These can be used with systemically administered bacteria rather than requiring intra-tumoural injection to expand the cancer types and tumour locations potentially amenable to treatment. For example, Chien et al. explored the growth of genetically modified EcN stimulated by a range of environments, such as hypoxia, altered pH and lactate, all known tumour microenvironment characteristics [57]. Once at the tumour site, payload release can also then be tailored to the tumour environment through the use of a genetic, synchronised lysis circuit that senses when bacteria reach quorum, achieved at high bacterial concentration, to activate programmed cell lysis and release of therapeutic payloads in tumours [34,58,59]. While not yet tested in humans, this strategy has been shown to reduce neoplastic burden following oral or intravenous administration of engineered bacteria to subcutaneous lymphoma and orthotopic colorectal cancer mouse models without generating systemic toxicities [8,34,58]. 

Further clinical development of engineered EcN strains for cancer treatment will also need to address the potential for EcN to produce the genotoxin colibactin. As the active agent in the probiotic Mutaflor^TM^, EcN does not pose any immediate safety concerns. However, the longer-term effects of EcN exposure in otherwise healthy individuals may be of concern. Similar to other B2 phylogroup *E. coli*, the EcN genome naturally encodes a polyketide synthase (*pks*) island that generates colibactin. EcN has been reported to be less genotoxic than other pks+ *E.coli* strains following co-culture with normal human or murine colon organoids, yet still demonstrably generated epithelial DNA damage and a colibactin-associated mutational signature [60,61]. Oral dosing of germ-free mice with wild-type *pks*^+^ EcN also caused DNA damage to epithelial cells lining the gut in vivo [62], and recently, the colibactin mutational DNA signature has been identified in approximately 12% of human colorectal cancer samples [60]. This suggests that exposure to *pks*^+^ bacteria, such as EcN, may cause DNA damage and cancer [62,63,64]. Promisingly, the probiotic and mutagenic properties of the *pks* locus may be able to be de-coupled to overcome such safety concerns [8,65,66,67]. We next consider what we know about why EcN homes to and colonises tumours, what dictates colonisation, and where exactly EcN localises within tumour tissues in order to understand how EcN chassis strains may be further optimised for cancer treatment.

## 5. How Does EcN Colonise Tumour Tissue?

### 5.1. Influence of Route of Bacterial Administration on Tumour Colonisation: Evidence from Pre-Clinical Studies

Different primary tumour locations and methods of bacterial administration have been utilised when studying the colonisation of EcN in mouse models to date (Figure 2). How tumour location and route of administration affect tumour colonisation by EcN may provide some clues about how EcN survives and travels to the tumour site. Direct comparison of intraperitoneal, intravenous or intra-tumoural EcN dosing to a subcutaneous mouse breast cancer model resulted in significant enrichment of EcN in tumours above off-target tissues, as measured by colony forming unit (CFU) assays with no difference between treatment groups [10]. For oral dosing, mice are generally administered EcN without an exogenous coating for enteric protection and at higher levels (10^8^–10^10^ CFU) [35,42,58,68,69,70] than is tolerated for intravenous, intraperitoneal or intra-tumoural administration (10^5^–10^7^ CFU) [12,32,33,34,35,38,39,50]. Oral dosing of orthotopic colorectal cancer mouse models resulted in significant colonisation of neoplastic lesions in the gut, both benign adenomas and bona fide adenocarcinomas, as well as liver metastases, in comparison to off-target control tissues [35]. This suggested that EcN can locally colonise gastrointestinal tract neoplasia, as well as travel to the liver from the gut [8,34,58]. Of note, however, is that oral delivery of EcN to a subcutaneous colorectal cancer xenograft model did not result in tumour colonisation, yet intravenous EcN treatment of the same mouse model did [35]. While oral dosing of subcutaneous mouse tumour models is not frequently reported, this suggests that EcN transit out of the gut and colonisation of subcutaneous tumours may be limited. However, this requires further head-to-head testing using a range of tumour models and a direct comparison of EcN administration routes. This is consistent with lower levels of bacterial colonisation of colorectal orthotopic and subcutaneous tumours observed after oral, compared to intravenous administration of *Fusobacterium nucleatum* (*F. nucleatum*) and *S. typhimurium* [71,72]. In summary, EcN is a well-validated coloniser of neoplastic tissue after local, peritoneal or intravenous administration. Incorporation of an enteric coating to protect against the harsh acidic environment of the stomach, as used in a mouse model of inflammatory bowel disease [73] and as found in Mutaflor^TM^ for human consumption, could improve tumour colonisation after oral dosing.

### 5.2. Learning from Host Infection by Enteric Pathogens

The exact mechanisms used by EcN to travel to tumour tissue are yet to be elucidated, but possible scenarios can be gleaned from known infection routes of related, pathogenic *E. coli* commensals. EcN is genetically most closely related to uropathogenic *E. coli* (UPEC), sharing 87% gene homology with UPEC strains CFT073 and ABU 83972 [74], yet EcN lacks typical virulence factors found in UPEC [75]. The mechanisms driving invasive infections of the urinary tract by UPEC have been reviewed [76]. In brief, UPEC can attach to the uroepithelium through Type I fimbrial adhesion moderated by the FimH adhesin [77], and this attachment instigates actin rearrangement leading to the envelopment and internalisation of UPEC within the host cell [77]. The Type I pili expressed by UPEC promote bacterial aggregation and generation of protective UPEC biofilms to escape immune clearance [78], leading to persistence in the bladder. Of note, EcN can also express Type I fimbriae and encode a *FimH* homolog [79]. However, the role of adhesins in EcN tumour colonisation has not been examined. Alternately, adherent-invasive *E. coli* (AIEC) strains that are pathogenic gut commensals adhere to intestinal epithelial cells (IECs) and translocate across enterocytes, as recently reviewed [80]. In brief, AIEC degrades the mucus barrier through the production of vat-AIEC mucinase, allowing for bacterial penetration to IECs [81]. A Type 1 pili variant, expressed on the outer membrane of AIEC associated with Crohn’s disease, attaches to IECs via recognition of carcinoembryonic antigen-related cell adhesion molecule six [82]. This results in tight junction reorganisation and loss of barrier integrity that enables AIEC to translocate [83,84]. Utilising a similar mechanism, AIEC can also bind to microfold cells (M cells) that reside close to lymphatic tissues [85], allowing for further translocation and macrophage invasion, exacerbating intestinal barrier injury [86]. The tumour-homing pathogen, *S. typhimurium,* also crosses M cells for dendritic cell uptake into intestinal lymphoid tissue, allowing for systemic dissemination [87]. Alternatively, oncogenic *F. nucleatum* may originate from the oral cavity and travel systemically through the bloodstream to colonise colorectal tumours, via recognition by bacterial Fap2 adhesion protein of N-Acetyl-D-galactosamine overexpressed on tumour cells [71,88]. This was supported by significantly higher tumour colonisation following intravenous bacterial injection, in comparison to repeated oral dosing, in orthotopic colorectal cancer mouse models [71]. 

It is likely that the mechanism of EcN tumour colonisation would differ from these exemplar species as it is a non-pathogenic, relatively non-invasive gut commensal bacterium. Conceptually, after oral dosing, the movement of EcN from the gastrointestinal tract to tumours requires survival in the gut, followed by translocation across the mucosal barrier and gastrointestinal tissue, as proposed for other tumour-homing bacteria [89] (Figure 3). The ability of bacteria to move across the mucosal barrier could be through the production of mucin-degrading mucinase [81], as previously discussed, or, in the setting of colorectal cancer, the mucosal barrier may be degraded [90]. Once at IEC, EcN could utilise Type 1 pili, F1C fimbriae, and/or components of the K5 capsule previously shown to be important for EcN epithelial attachment [44,91], and move transcellularly through enterocytes (Figure 3), in a similar manner to UPEC and AIEC [82]. Intra-luminal tumours in contact with the gut microbiome may be colonised by EcN as part of a surface or intra-tumoural biofilm [8] (Figure 4). Once across the gut barrier, EcN may move into the blood stream via the mesenteric lymphatic system, as shown for *E. coli* gut commensals [92], or following phagocytosis by immune cells, as previously suggested to occur in vivo [30] (Figure 3). Further research is required to precisely understand the mechanism of EcN tumour homing and colonisation, including a complete assessment of the effect of differing locations, genetics and histopathologic subtypes of primary tumours on bacterial colonisation and in the setting of competing host and tumour microbiomes. This will inform the design of clinical testing for EcN-based therapeutics and may highlight those patients or cancer subtypes in which these strategies may be most effective.

## 6. Inherent Tumour Properties Underlying Colonisation by Bacteria

### 6.1. Aberrant Tumour Vasculature and Inflammation Permit Bacterial Entry

Two key tumour intrinsic factors proposed to underlie colonisation by bacteria are; (1) the chaotic, slow flowing and leaky tumour vasculature that entraps haematogenous bacteria within the tumour environment [93] and (2) tumour-associated inflammation that floods trapped bacteria into the tissue [15]. As definitive experimental data describing EcN entry to tumour tissue from the vasculature are lacking, we focus here on mechanisms described for other tumour-resident bacterial species. Forbes et al. visualised the in vivo entrapment of GFP-expressing *S. typhimurium* within the tumour vasculature of a subcutaneous mammary carcinoma mouse model. Intravital microscopy showed *S. typhimurium* adhering to vessels with a lower flow rate, similar to that of the tumour vasculature, in contrast to high flow rate vessels where no bacteria adhered [93] (Figure 3). Furthermore, observation of histological samples taken at multiple time points after *S. typhimurium* infection supports the tumour vasculature as a route of tissue entry. Bacteria were initially present near the vasculature before disseminating further into the tumour tissue [16,18] (Figure 3). Contributing to bacterial entry from the tumour vasculature could be an inflammatory influx, as observed in a subcutaneous murine colorectal cancer model [15]. Histological observations suggested a concurrent red blood cell influx and increased pro-inflammatory cytokine production, particularly Tumour Necrosis Factor (TNF)-α, that washed bacteria into the tumours from the vasculature [15] (Figure 3). Of note, prior to this inflammatory influx, some bacteria had already escaped the tumour vascular and were located around the tumour blood vessels, suggesting an additional, alternate route for bacterial movement into tumours, such as increased vascular permeability as described for the movement of *F. nucleatum* from the oral cavity to colorectal tumours [71]. 

### 6.2. Variable Oxygen Supply in Tumour Tissue

Based on metagenomic sequencing of bacteria in human tumour samples, both anaerobic and aerobic bacteria are naturally present within the tumour microbiome [94]. It is evident that, to date, research investigating exogenous tumour colonising bacterial strains in pre-clinical cancer models has predominately utilised obligate anaerobes (bacteria that have limited survival in the presence of oxygen, such as *Fusobacterium* [71,88] and *Clostridium* [13]) and facultative anaerobes (including *E. coli* [6,7,8,9,10,11,12] and *Salmonella* [14,15,16,17,18]). A limited number of studies reported successful tumour colonisation by aerobic bacteria, such as *Bacillus subtilis* and *Pseudomonas aeruginosa* [95]. As such, how the fastidious oxygen requirements of specific bacteria, or flexibility in varied oxygenation states as found in EcN, then influences tumour colonisation remains an open question in the field.

### 6.3. Tumour Necrosis—The Ultimate Metabolic Niche for Bacteria?

Regions of cell death, or necrosis, are often observed within solid tumours, as the requirements of the growing tumour outstrip the ability of the tumour neovasculature to supply oxygen and nutrients. These necrotic cores usually form in hypoxic regions, providing an ecological niche for fastidious and facultative anaerobes, and metabolites for bacterial growth. Across numerous subcutaneous tumour mouse models of cancer following intravenous or intra-tumoural administration of different bacteria, including *S. typhimurium*, EcN, *E. coli* TOP10 and K-12, *Shigella flexneri* (*S. flexneri*), *Clostridium beijerinckii* and *Bifidobacterium longum*, there is a consistent pattern of bacteria localising within and predominately around the rim of the necrotic core, with very few to no bacteria present in viable tumour tissue as identified on histopathological analyses, acknowledging the varying degrees of histological resolution in these studies based on available imaging technology at the time and specificity of stains used to visualise the bacteria [10,12,13,15,17,18,28,93,96,97]. Only the administration of *S. flexneri* generated observable but very few bacteria present within the viable tumour region. However, this was suggested to be due to the intra-tumoural bacterial delivery method rather than any preferential localisation to this region [97]. Three studies have reported that smaller subcutaneous tumours or cylindroid models that lack necrotic cores, both in vitro and in vivo, were not effectively colonised by bacteria [93,95,96]. However, these studies lack detailed information to understand how size and/or the presence of a necrotic core impacts colonisation ability. 

To gain a picture of why and how bacteria distribute within tumours, researchers have focused on understanding bacterial movement to the necrotic core over time. Timed endpoint studies analysing the accumulation pattern of *S. typhimurium* after intravenous dosing to a subcutaneous mammary carcinoma mouse model have shown the early accumulation of bacteria around the tumour edge, close to the vasculature, at 12 h [15,18]. By 48 h, *S. typhimurium* disseminated throughout the whole tumour before accumulation in the transition zone, the area between viable and necrotic tumour tissue, and reaching the necrotic tumour core by 96 h [15,18]. Similar data were reported in the same model for the *E. coli* strain K-12, where patchy clusters of bacteria were visualised throughout the whole tumour at 24 h post-intravenous administration before settling within the transition zone by 72 h [98]. Detailed temporal studies have not yet been reported for EcN colonisation of solid tumours. However, why EcN localises preferentially within the partially oxygenated transition zone may be gleaned from in vitro studies using a microfluidic chamber to control environmental oxygen [99]. These studies showed EcN grew more efficiently in oxygenated environments over anoxic, such as would be found within the necrotic core [99]. The presence of *E. coli*, specifically EcN [12] and K-12 [98], in the tumour transition zone also reportedly led to increased necrosis [12,98], possibly as a result of upregulation of TNF-α leading to tumour cell apoptosis and decreased blood vessels within the region [98]. This vascular remodelling may underlie the relatively large distances, on average 360 μm to 750 μm, visualised between a tumour resident bacterium, in this case, *S. typhimurium,* and blood vessels within the tumour [16,93]. Another study suggested that expanding the necrotic region with vascular disrupting agents prior to *S. typhimurium* dosing increased the ability of the bacterium to target and colonise tumours [100].

Chemotaxis describes the migration of cells in response to chemical stimuli in the environment. As EcN is of the H1 serotype, it is motile and can move towards stimuli using flagella [21]. Interestingly, while both *S. typhimurium* and EcN flagella mutant strains showed significantly decreased motility in vitro, intravenous administration to subcutaneous mouse models of breast and colorectal cancer resulted in the same degree of tumour colonisation as wild-type strain comparators in vivo [29,72]. This suggests that flagella may be unnecessary for tumour colonisation after venous delivery. *S. typhimurium*, at least, may instead utilise known systemic infection routes via endothelial and phagocytic cells [87], as previously mentioned. Metabolites preferentially utilised by EcN in vitro [21] and in the intestinal niche [101] have been identified, such as galactose, fucose, arabinose, arginine and dextrose [21,101]. However, little of this research has extended to cancer-bearing models to date. Understanding in vivo tumour metabolite use by EcN could inform the design of auxotrophic and tumour microenvironment-regulated EcN strains for optimal cancer diagnostics and therapeutics. 

The importance of chemoreceptors for *S. typhimurium*, but not EcN, motility within the tumour microenvironment has been modelled in vitro using a colorectal cancer cell cylindroid system. In this model, movement of *S. typhimurium* to the cylindroid edge required an aspartate receptor, penetration of the viable tumour cell region required a serine receptor, and chemotaxis towards regions of necrosis required a ribose/galactose receptor [102]. These experiments illustrated the plasticity of bacteria to adapt to the varied metabolic niches present within a tumour-like environment. Indeed, mutation of individual glucose or ribose receptors or perturbation of aromatic amino acid biosynthesis in *S. typhimurium* did not alter the number of bacteria that colonise subcutaneous breast or colorectal tumours in mice after intravenous injection [16,72]. This suggests that attenuation of specific metabolic pathways can be compensated for by the bacterium within the tumour microenvironment, enabling tumour colonisation to proceed. It was noted, however, that whilst the mutant and wild-type bacteria localised within similar tumour regions in vivo, the mutant strains had a 2-fold higher ability to invade, as demonstrated by increased distance from vessels [16]. Preliminary investigations examining the reliance of EcN on specific chemoreceptors for tumour colonisation require additional follow-up. Intravenous administration of aromatic amino acid (*aroA*) mutant EcN to a subcutaneous breast cancer model resulted in a 200-fold reduction in tumour colonisation in comparison to wild-type EcN [29]. This would suggest that the aromatic amino acid pathway may be key to EcN tumour colonisation. However, the authors did not report whether the *aroA* mutation also affected in vitro EcN growth, but there were also significantly lower CFU counts of the mutant EcN in off-target organs, suggesting overall bacterial fitness (separate from colonisation potential) may have been affected by the *aroA* mutation. 

When taken together, these studies reinforce the idea that bacteria use a combination of mechanisms, including altered tumour vasculature, inflammation and metabolic niches, to colonise tumours. Further research is required to detail specific mechanisms used by EcN for tumour colonisation. 

### 6.4. Aberrant Immune Surveillance in the Tumour Microenvironment

The presence of bacteria in healthy people can generate an immune response, as reviewed [103]. Briefly, activation of the immune system occurs through pattern recognition receptors (PRRs) that identify microbes through pathogen/microbe-associated molecular patterns [103]. Activation of PRRs can trigger innate immune responses that upregulate pro-inflammatory cytokines, such as Interleukin (IL)-1β, IL-18, IL-6, TNF-α and Type I and III IFN, ultimately activating macrophages and neutrophils [103]. If uncleared, the chronic inflammation and microbial presence activate the adaptive immune system, where particular T cells respond, depending on whether the bacteria reside intracellularly, such as TH_1_ cells, or extracellularly, such as TH_17_ cells [103]. The immunophenotype of the tumour microenvironment can vary greatly as tumours evolve to escape regulation by the immune system, a process known as immunoediting. This immunosuppressed tumour tissue can be an ideal, immune protected niche perfect for bacterial colonisation.

Two studies have experimentally manipulated the innate immune system prior to intravenous administration of bacteria in tumour-bearing mice to investigate the role of the immune system in preventing tumour colonisation by bacteria (not limited to EcN). Either clodronate-containing liposomes (CLIP) were used to deplete macrophages, or anti-Gr1 antibodies to deplete neutrophils. Systemic treatment of a syngeneic, orthotopic mouse breast cancer model with CLIP did not affect tumour colonisation by *S. typhimurium* but significantly increased EcN tumour and spleen colonisation in comparison to controls [29]. This suggests that EcN colonisation of tumour and off-target tissue is limited by host macrophages. Similarly, systemic treatment of a syngeneic, subcutaneous mouse colon cancer model with anti-Gr1 resulted in significantly higher colonisation of tumour tissue and off-target organs after intravenous administration of *S. typhimurium* or the non-pathogenic *E. coli* strain TOP10 [97]. This, unsurprisingly, suggests that neutrophils play a key role in defence against tissue colonisation by bacteria, including the tumour microenvironment. Alternatively, the role of the adaptive immune system in preventing tumour colonisation by bacteria has been investigated after intravenous or intra-tumoural administration of EcN or *E. coli* K-12 to immunocompetent mice bearing subcutaneous mammary carcinomas in comparison to matched, immunocompromised controls that lack T cells [10,19,98]. Both *E. coli* strains colonised tumours to the same degree in all mouse models [10,98], indicating that tumour colonisation by EcN is preferentially attenuated by the innate, rather than the adaptive immune system. In contrast, oral treatment of a similar mouse mammary tumour model with *Bifidobacterium breve* resulted in 3-fold higher tumour colonisation in T-cell deficient *nu*/*nu* mice, in comparison to immunocompetent animals [19]. This suggests that the presence of T-cell adaptive immunity differentially impacts colonisation for different bacterial strains. This warrants validation in further studies across a wider range of tumour sites, types and direct comparison of methods of administration, with a focus on EcN.

At the cellular level, the distribution pattern of bacteria within tumours is likely to be influenced by the spatial distribution of immune cells and vice versa. In human colorectal and oral squamous cell carcinoma samples, the native tumour microbiome, located using both spatial transcriptomics and high-resolution co-staining experiments, was found to reside in discrete, immunosuppressive niches, characterised by an increase in CD66b^+^ [104], a marker for granulocytes, including neutrophils. Consistent with this, in mouse breast, melanoma [12] and colorectal cancer [15,97] models, neutrophils have been frequently reported to encircle bacteria that reside in the transition zone and necrotic tumour core [12,15,97] from 18 h post-administration of EcN [12], *S. flexneri* [97] or *S. typhimurium* [15,97]. Systemic neutrophil depletion led to observable bacteria within viable tumour tissue by pathological analysis, demonstrating that neutrophils likely limit *S. typhimurium* and *S. flexneri* distribution within tumours [97], with similar analysis with EcN not undertaken. Mechanistically, bacterial colonisation upregulates tumour TNF-α expression [15,30,98], which acts as an attractant for neutrophils [15]. 

Macrophages, similar to neutrophils, have also been shown to localise between viable tumour tissue and the transition zone to necrosis, residing close to phagocytosing bacteria as visualised by intracellular staining of *E. coli* K-12 in a subcutaneous mammary carcinoma mouse model [98]. Macrophages are a likely source of TNF-α within the tumour microenvironment [98], as supported by the upregulation of genes involved in TNF-α and IL-1β production in macrophages that had engulfed native bacteria in patient colorectal and oral squamous cell carcinoma samples [104]. As well as macrophages, IL-1β production by dendritic cells within the tumour microenvironment was also shown to be induced by intravenous *S. typhimurium* treatment in a subcutaneous colorectal cancer model [14]. However, interestingly, in this model, the same IL-1β and TNF-α response was not observed after intravenous delivery of *E. coli* MG1655 [14], suggesting further investigation is required to understand host responses to different bacterial species, including EcN. While high-definition, spatial analysis of the tumour microbiome and immune compartment in patient samples is still in its infancy, there is some overlap between reports from cancer patients and mechanistic mouse studies covered here. The specific, high-resolution temporal and spatial localisation of EcN and host immune cells within tumours in vivo has not yet been elucidated. Understanding the co-regulation of bacterial and immune compartments may lead to improved tumour targeting and therapeutic payload release specific to immune components by engineered EcN in the future, as exemplified by the Synlogic approach described above that utilises host phagocytic pathways to activate the STING pathway and prevent tumourigenesis.

‘Stealth’ EcN, EcN coated with erythrocyte cell membranes or increased capsular polysaccharides (CAPs), have recently been employed to modify the influence the immune system has on bacterial tumour colonisation [33,38]. In comparison to uncoated EcN, administration of erythrocyte-coated EcN generated a significant reduction in inflammatory immune response markers, IL-6, IL-10 and TNF-α, after intravenous injection into BALB/c mice [38]. Similarly, EcN expressing increased CAPs reduced Il-1β, IL-10 and granulocyte colony-stimulating factor in comparison to uncoated EcN [33]. Both types of ‘stealth’ bacteria showed higher levels of tumour colonisation whilst also maintaining clearance from off-target organs [33,38]. These examples illustrate how the immune system may be circumvented to optimise bacterial colonisation and localisation within the tumour.

## 7. Conclusions and Future Directions

Tumour-homing bacteria present an exciting cancer treatment tool to overcome the current therapeutic challenges inherent to the low-oxygen and immune-privileged tumour microenvironment. Of the candidate tumour-homing bacteria studied to date, EcN provides many favourable characteristics for translational use, including oxygen tolerance and low pathogenicity, coupled with a well-characterised genome and ease of genetic manipulation for synthetic biology approaches. EcN also has a history of probiotic use in people, with the recently closed Phase I Synlogic clinical trial demonstrating the safety of genetically modified EcN in cancer patients. Furthermore, pre-clinical research utilising genetically modified EcN has consistently shown cancer therapeutic success with tumour-specific targeting, low background in off-target tissues and lack of systemic toxicities.

This review outlines potential mechanistic explanations for the distribution and localisation of EcN within tumours, when possible, through a summary of existing EcN literature while also addressing the knowledge base more broadly for other tumour-homing bacteria. Key factors that may regulate tumour homing include avoidance of immune clearance, altered tumour vasculature and bacterial adaptability to changes in oxygen levels. However, there is limited mechanistic data for EcN. The field would benefit from detailed functional analyses, for example the use of EcN fimbriae and Type 1 pili mutant strains and assessment of effect on tumour colonisation in vivo. What is better characterised with respect to the intra-tumoural localisation of EcN and various other bacteria is the preferential accumulation within the transition zone and necrotic core. However, further work is required in this area to understand how the bacteria distribute and move to this region of the tumour, information that would prove highly valuable if understood. Future studies require high-quality resolution outputs, like electron and confocal microscopy, high-definition spatial transcriptomics or in situ hybridisation techniques such as RNAscope [8,104], with intravital imaging or temporal assessments post-bacterial administration to best characterise bacterial distribution and localisation patterns. Important design considerations for engineered bacteria for cancer treatment, including EcN, have been covered previously [105,106] and include biocontainment of genetically modified bacteria, such as the incorporation of auxotrophic mutations with reduced fitness outside the tumour microenvironment; careful evaluation of exogenous DNA sequences, including and stability of genetic elements; removal of antibiotic selection cassettes to reduce risk of increased antibiotic resistance; deletion of potentially pathogenic features such as colibactin production. Given the rapid and ongoing increase in cancer studies utilising EcN as a bacterial chassis over the last decade, this information will enable the intelligent design of the next generation of tumour-specific payload delivery encoded or carried by the EcN chassis.

## Figures and Tables

**Figure 1 cancers-16-02971-f001:**
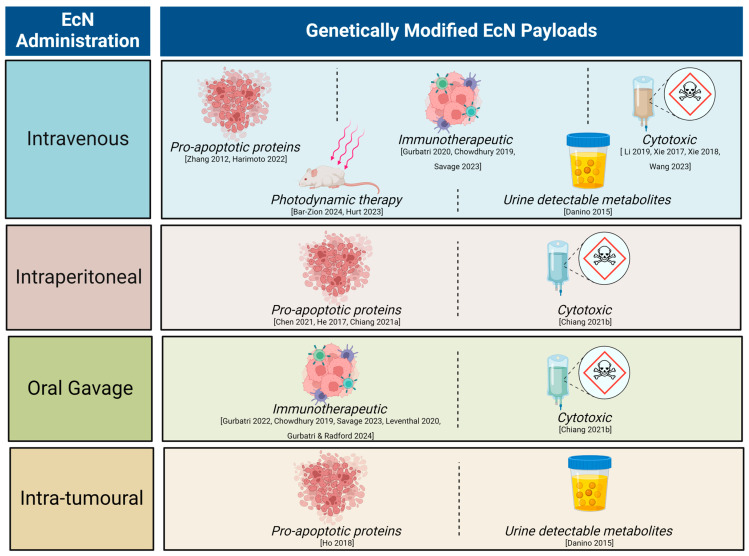
Summary of pre-clinical literature utilising genetically modified *Escherichia coli* Nissle 1917 (EcN) with different modes of administration. Recent reviews have covered the breadth of therapeutic molecules delivered by a range of engineered bacteria [48,49,50]. Here, we focus on the EcN chassis strain. Engineered strains of EcN have been successfully used to detect neoplasia or reduce tumour burden via genetic encoding to deliver a variety of payloads across many tumour types [7,8,9,11,12,30,31,33,34,35,36,37,40,42,51,52,53,54,55].

**Figure 2 cancers-16-02971-f002:**
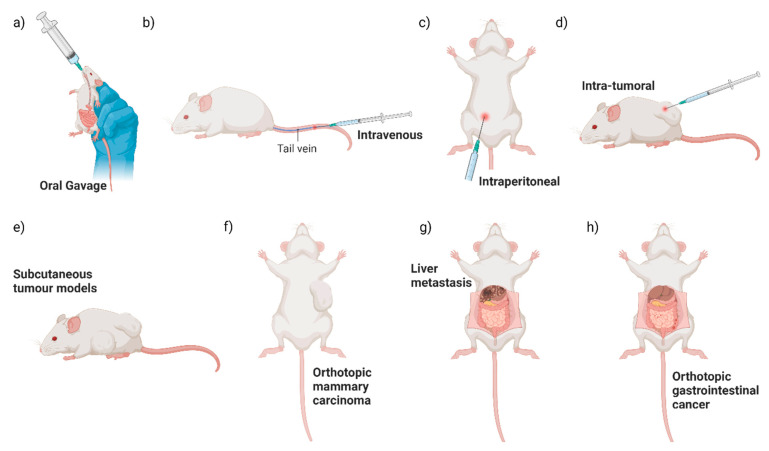
Summary of pre-clinical routes of administration and mouse tumour models. Pre-clinical testing of engineered EcN has commonly used a variety of routes of EcN administration (**a**–**d**) and mouse tumour models (**e**–**h**) illustrated here.

**Figure 3 cancers-16-02971-f003:**
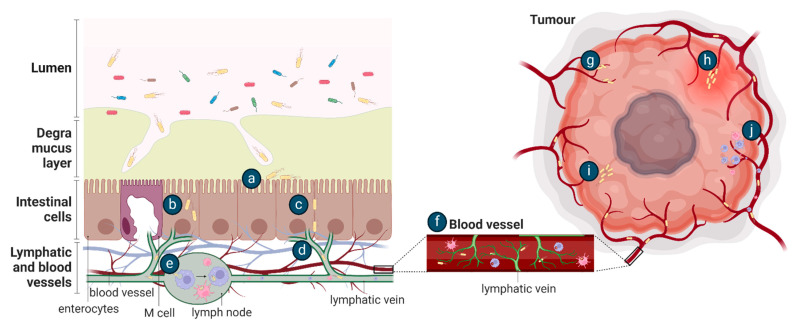
Potential methods of bacterial tumour colonisation after oral (**a**–**j**) or intravenous (**e**–**j**) administration. After oral delivery, EcN in the gut lumen may (**a**) adhere to the intestinal epithelial layer following translocation across the mucus layer and then be transported (**b**) transcellularly through enterocytes or (**c**) paracellularly through tight junctions and (**d**) move into the lymphatic system. EcN may be (**e**) phagocytosed and (**f**) travel within immune cells into the bloodstream or delivered directly to the blood by intravenous injection. EcN may then enter a tumour by (**g**) becoming entrapped in the chaotic tumour vasculature, (**h**) flooding into the tumour with inflammation, (**i**) by bacterial chemotaxis towards the tumour or by (**j**) the lysis of phagocytic host cell to release the bacteria.

**Figure 4 cancers-16-02971-f004:**
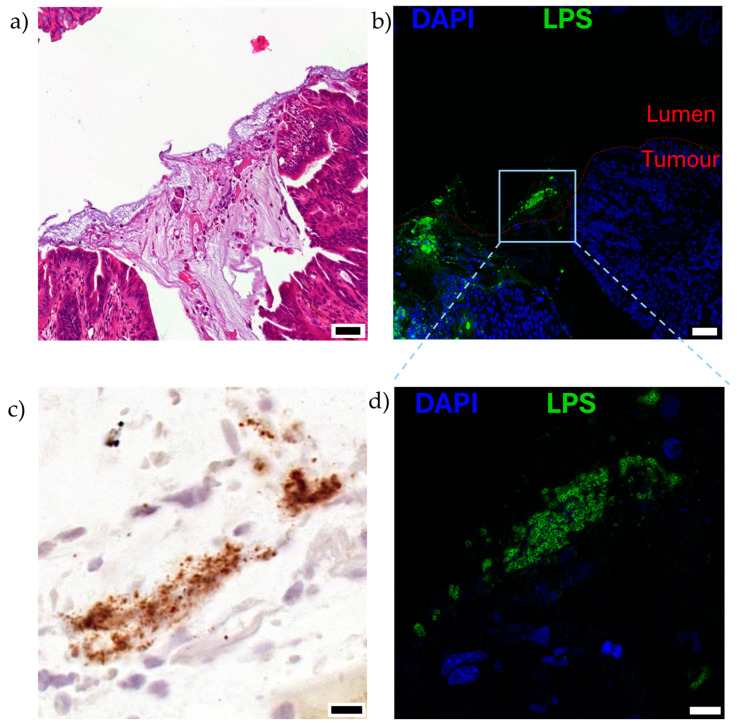
Visualisation of EcN on the luminal surface of mucinous, orthotopic colorectal tumours following oral dosing to mice, as described in [8]. Serial sections of the luminal surface of a representative orthotopic colorectal tumour are shown (n = 5). Mice bearing colorectal tumours located in the distal colon were orally gavaged twice with 10^10^ CFU EcN over 2 days and analysed 5 days after the last dose. Serial sections stained with (**a**) hematoxylin and eosin, mucinous region can be observed in darker purple, acellular region in the centre of the image, (**b**,**d**) immunofluorescence for lipopolysaccharide (LPS; green), a pan-Gram-negative bacterial stain, and nuclei (4′,6-diamidino-2-phenylindole, DAPI; blue), and (**c**) EcN specific location by RNAscope in situ hybridisation for luciferase (lux) cassette genetically engineered into bacteria (brown). Boxed region in (**b**) is shown at higher magnification in (**c**,**d**). Scale bars (**a**,**b**) 50 μm, (**c**,**d**) 10 μm.

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
