# Peer review of "Towards Understanding Tumour Colonisation by Probiotic Bacterium E. coli Nissle 1917"

_cancers, 2024, doi:10.3390/cancers16172971_

Round 1

Reviewer 1 Report

Comments and Suggestions for Authors

 The main idea of this Review is that genetically modified Escherichia coli Nissle 1917 (EcN) has shown promise as a tool for cancer treatment and detection due to its ability to selectively colonize tumors over normal tissues. Pre-clinical studies have demonstrated that EcN can effectively deliver therapeutic agents, reducing tumor burden with minimal off-target effects. Additionally, the EcN chassis has been expanded into a potential tumor detection tool. However, a detailed mechanistic understanding of how EcN homes to and localizes within tumors is still lacking, which is crucial for optimizing its use in cancer therapy and addressing safety concerns.

This is a very innovative Review providing exciting evidence. I also thank the Authors for perfect writing style.  The Review can be published. There are only several minor recommendations:

Abstract: To attract more citations, please, explain anti-cancer activity
of genetically engineered EcN in more details.
This information
is amazingly interesting. It also would be good to underline principal novelty of this Review and to explain novel analytical findings.

Conclusion: Please, outline directions of future investigations in more
details to create more structured plan for future research.
It also would be good to mention potential biosafety and ethical
concerns associated with using genetically modified bacteria in
cancer treatment, especially regarding long-term effects.
By addressing these concerns, the conclusion would present a more
balanced and comprehensive perspective on the current state
of research.

Author Response

We are grateful to the reviewer for their enthusiasm and comments that have 
helped us substantially improve our manuscript. Please see the attachment for point by point response.

Reviewer 2 Report

Comments and Suggestions for Authors

The paper, through a summary of existing EcN literature, indicated mechanistic explanations for the distribution and localisation of EcN within tumours, when possible. Key factors that may regulate tumour-homing were mentioned and commented on; however, there is limited mechanistic data for EcN.

There are some minor changes that need to be fixed

-        A graphical abstract associated with the paper will improve the impact to the readers

-        At the end of the introduction is important to add a short sentence mentioning the review strategy and the main directions that will be followed in the main part of the article. It cannot enter directly in Discussion..

-        Some comparative tables underlining the anti-cancer activity could be important to understand the similarities and differences between previous studies done

-        In order to explain “varying degrees of histological resolution in these studies[6,16,18,20,22,23,33,92,95,96]” it is recommended to include some more examples/comments..

-        Some words/expressions need to be more scientific – “This review seeks to summarise”,  “this review has sought”, “A handful of studies”…

-        For Figure 1, Figure 4 – d- some words are not completely indicated - DAPI

-        Space before the brackets indicating the reference.

Comments on the Quality of English Language

-        Some words/expressions need to be more scientific – “This review seeks to summarise”,  “this review has sought”, “A handful of studies”…

Author Response

(The authors gave the same response as above.)
